# Is There a Causal Link between the Left Lateralization of Language and Other Brain Asymmetries? A Review of Data Gathered in Patients with Focal Brain Lesions

**DOI:** 10.3390/brainsci11121644

**Published:** 2021-12-13

**Authors:** Guido Gainotti

**Affiliations:** Institute of Neurology, Catholic University, 00168 Rome, Italy; guido.gainotti@unicatt.it

**Keywords:** hemispheric asymmetries, language and cognition, emotional and cognitive systems, automatic and unconscious aspects of the right hemisphere functioning

## Abstract

This review evaluated if the hypothesis of a causal link between the left lateralization of language and other brain asymmetries could be supported by a careful review of data gathered in patients with unilateral brain lesions. In a short introduction a distinction was made between brain activities that could: (a) benefit from the shaping influences of language (such as the capacity to solve non-verbal cognitive tasks and the increased levels of consciousness and of intentionality); (b) be incompatible with the properties and the shaping activities of language (e.g., the relations between language and the automatic orienting of visual-spatial attention or between cognition and emotion) and (c) be more represented on the right hemisphere due to competition for cortical space. The correspondence between predictions based on the theoretical impact of language on other brain functions and data obtained in patients with lesions of the right and left hemisphere was then assessed. The reviewed data suggest that different kinds of hemispheric asymmetries observed in patients with unilateral brain lesions could be subsumed by common mechanisms, more or less directly linked to the left lateralization of language.

## 1. Introduction

The Broca’s discovery that a specific defect of language production is caused in humans by a lesion of the ventral part of the left frontal lobe [1,2] was prompted by the great interest raised in the scientific milieux by Darwin’s theory on the origins of species [3] and contributed substantially to the development of the principles of functional localization and of hemispheric specialization. Furthermore, since language is considered a hallmark of our species, for more than one century hemispheric asymmetries were considered as a uniquely human feature. In more recent times, however, this idea has been challenged by an increasing number of studies describing hemispheric asymmetries for various behavioural functions in many non-human species (see reviews in [4,5,6,7]). Several theories have therefore been advanced to explain the general advantages and disadvantages of hemispheric specialization, mainly suggesting that this division of work could reduce the need for processing information in both hemispheres and the need for interhemispheric communication via the corpus callosum. Thus, reviewing animal studies, Vallortigara and Rogers [6] have suggested that a complementary specialization of functions is advantageous when carrying out simultaneous processing and that this specialization may have contributed to the evolution of cognitive lateralization. Analogously, Kosslyn [8], making reference to human studies, postulated that the two critical neural control systems, dealing respectively with speech and with shifts of spatial attention, work better if they are controlled by different cerebral hemispheres. This model has been considered as a “causal hypothesis” and is consistent with the so-called “crowding model”, which states that, to avoid interference, spatial attention performance will be crowded out from the language-controlling hemisphere. The alternative model of the human hemispheric asymmetries is the “statistical hypothesis”, which maintains that complementary specialization is a statistical (rather than a casual) phenomenon and that different neural functions lateralize independently of each other. The causal hypothesis assumes that a relation may exist between different aspects of hemispheric asymmetries (e.g., the lateralization of one function causes the opposite lateralization of the other) and that this relation may derive from a longstanding evolutionary origin. On the contrary, the statistical hypothesis assumes that different functions may lateralize independently and that no link exists between different kinds of hemispheric lateralization [9]. Therefore, if one could show that a relation exists between different aspects of brain lateralization, this should support the causal hypothesis. Until now, however, little empirical evidence has been provided to support either the causal or the statistical hypothesis (reviewed by Bryden, [10]) in humans and all these empirical studies have been conducted in groups of healthy human volunteers. Zickert et al. [11] recently noted that only in very few of these studies participants actually performed two tasks simultaneously (as postulated by the “crowding hypothesis,”) and examined a possible positive effect of the strength and direction of lateralization on two demanding cognitive tasks. Their results did not support a crowding effect, because the strength of lateralization predicted the performance measure of both tasks, but no association between direction or strength of lateralization was found for the dual task. Furthermore, one may wonder whether results obtained on a specific experimental investigation can necessarily be generalized to the solution of a complex general problem. The aim of the present paper, therefore, consisted of trying to investigate if some support for the causal hypothesis could be obtained by a careful review of data gathered in patients with unilateral brain lesions, rather than by an experimental investigation in healthy subjects. The rationale of this approach was twofold:On one hand, the view, largely shared since the Broca’s discovery, that language may have a top-down influence on various aspects of cognition and may thus shape general aspects of the brain functioning (e.g., [12,13,14,15,16]. It is interesting on this subject to note that Premack [17] has shown that chimpanzees who receive language training have superior reasoning and problem-solving skills compared to language-naïve chimpanzees, supporting the notion that representational language facilitates many kinds of cognitive abilities.On the other hand, the uncontroversial notion that the study of patients with unilateral brain lesions has given a fundamental contribution to the development of our knowledge about hemispheric asymmetries in humans.

The structure of the present review will, therefore, consist of three parts:(1)A short introductory section in which an attempt will be made to identify three different kinds of brain activities: (a) the (mainly cognitive) activities that could benefit from the shaping influences of human language and should, therefore, be affected by a left hemisphere (LH) damage; (b) the brain functions that are incompatible with the properties of language (and should, therefore, be subsumed by the right hemisphere (RH) to avoid interferences); (c) the brain activities that could be represented more on the right than on the left hemisphere, due to the competition for cortical space determined by the development of language within the LH, rather than to the functional properties of language.(2)A more sizeable section in which attention will be shifted from the hypothesized influence of human language on different kinds of brain activities to the neuropsychological data gathered in patients with unilateral brain lesions. The main purpose of this section will consist in checking the correspondence between predictions based on the theoretical impact of language on other brain functions and data obtained in patients with lesions of the right and left hemisphere.(3)A short conclusive section in which an attempt will be made to evaluate if these data support or not the general hypothesis assuming that the left lateralization of language may have a causal link with other (apparently non-language-related) hemispheric asymmetries.

## 2. Identification of Brain Activities That Should Theoretically

(a) benefit from the shaping influence of language; (b) be subsumed by the right hemisphere to avoid interferences with language activities; (c) be more represented on the right hemisphere due to competition for cortical space.

### 2.1. Brain Activities That Could Theoretically Benefit from the Use of Language

From the theoretical point of view, different kinds of cognitive activities should take advantage of the use of language and could, therefore, be negatively affected by left hemisphere damage. Furthermore, Jackendoff [14] has rightly noted that language can influence not only the content but also the structure of thought. Among the cognitive abilities that could be affected by language disorders, the most important ones from the present viewpoint could be those involved in non-verbal cognitive tasks. On the other hand, the more general aspects of thought that could be shaped by language could mainly concern the levels of consciousness and of intentionality. The cognitive skills that will be taken separately into account in the analysis of data gathered in patients with unilateral brain lesions will, therefore, concern the capacity of aphasic patients to solve non-verbal cognitive tasks, whereas the levels of consciousness and of intentionality will be considered as the more general aspects of thought that could be influenced by the left lateralization of language.

#### 2.1.1. The Capacity to Solve Non-Verbal Cognitive Tasks

This capacity can be facilitated by the use of an intermediate verbal mediation between the initial conception and the final execution of the act, as studies conducted in children have shown that solving verbal and non-verbal tasks is associated with the use of inner speech [12,18,19]. It must, however, be acknowledged that many different abilities are requested to solve non-verbal cognitive tasks. In the section dealing with the relationships between language disorders and non-verbal cognitive defects, a clear distinction should, therefore, be made between the non-verbal cognitive tasks that are significantly impaired and those that are not impaired in aphasic patients with left-brain lesions.

#### 2.1.2. More General Aspects of Thought (Such as the Levels of Consciousness and of Intentionality) That Could Be Influenced by the Left Lateralization of Language

Many classical scholars (e.g., [14,20,21]) argue from different perspectives that language use enables consciousness. According to Jackendoff [13], inner speech helps to pay attention to relational and abstract aspects of thought and this fact allows us to clarify the relations among language, thought and awareness. In a similar vein, Searle [20] maintains that much of our consciousness is shaped by language. Furthermore, Arbib [21] recalls Cajal’s view that human consciousness may share aspects of “animal awareness” with other non-human species but has its unique form because humans possess language. Following the same line of thought, several authors maintain that language shapes not only consciousness but also the highest levels of intentional/controlled activities. Thus, Savage-Rumbaugh et al. [22] acknowledge that intentionality has a language-related dynamic quality and Brandom [23] makes a distinction between practical and discursive intentionality. The former is a sort of ‘directedness at objects’ that animals exhibit when they deal skillfully with their world, whereas the latter is only exhibited by concept-users in the richest sense. The relations between levels of intentionality and of consciousness are highlighted by the interactions documented between the dichotomies ‘automatic vs. controlled/intentional’ and ‘unconscious vs. conscious’ processes. Both in perceptual recognition (e.g., [24]) and in action control (e.g., [25]), intentional/controlled” processing tends, in fact, to be associated with consciousness, whereas “automatic” processing tends to be associated with unconscious mechanisms.

### 2.2. Brain Functions That Could Be Subsumed by the Right Hemisphere to Avoid Interferences with Language Activities

Two main brain systems could theoretically be subsumed by the right hemisphere to avoid interferences with language activities: (a) the emotional system and (b) the visual-spatial attention orienting system.

#### 2.2.1. The Relations between Language and Emotions

Interferences between language and emotions should be avoided because language plays an important role in cognitive activities and (from the theoretical point of view) the emotional and the cognitive system are considered as two alternative adaptive systems used to interact with a partially unpredictable environment [26,27,28,29,30,31]. From this point of view, the emotional system is regarded as an emergency system, able to interrupt ongoing activity to rapidly select a new operative scheme, whereas the cognitive system is viewed as a more complex conscious and controlled system, which requires more time to carry out its work. Conflicts could, therefore emerge between the analysis of the sensory data and the selection of the behavioral response accomplished by the emotional and by the language-related adaptive cognitive system.

#### 2.2.2. The Relations between Language and Visual-Spatial Attention Orienting System

As already mentioned in the Introduction of this paper, Kosslyn [8] has postulated that in humans two main neural control systems, dealing respectively with speech and with shifts of spatial attention, are controlled by different cerebral hemispheres. This author assumed that a categorical speech control subsystem, located in the LH, should produce representations that correspond to those stored in the associative memory of this hemisphere. On the other hand, the RH should play a special role in a control system dealing with the rapid shifts of attention over a scene. This visual-spatial attention control system should be distinguished from a wide range of other attentional functions that need not be lateralized, in the same way. Another reason which could suggest a link between automatic orienting of attention and RH structures is the capture of attention by possible sources of threat because this capture of attention is the first step of the sequence of events that leads to the generation of emotions. Therefore, if emotions are mainly subsumed by RH structures, it should be logical to expect that automatic orienting of attention may also be linked to the same RH structures.

### 2.3. Brain Activities That Could Be More Represented on the Right Than on the Left Hemisphere Due to Competition for Cortical Space

The mechanism of competition for cortical space [32] could imply that, when language developed in the LH, areas homologous to those involved in language processing in this hemisphere remained dedicated to more basic sensory-motor functions in the right hemisphere. This hypothetic mechanism has been suggested by Gazzaniga [33] with the following words: “While language emerged in the left hemisphere at the cost of pre-existing perceptual systems, the critical features of the bilaterally present perceptual system were spared in the opposite half-brain”. It is, therefore, possible that, besides the asymmetries due to the shaping influence of language on the cognitive activities and to the negative interferences between language and other brain functions, a further asymmetry between the two halves of the brain may be due to the competition for cortical space.

### 2.4. Results Obtained in Patients with Unilateral Brain Lesions That Could Support the Hypothesis of a Causal Link between the Left Lateralization of Language and Other Brain Asymmetries

On the basis of considerations developed in previous paragraphs of this section, it could be predicted that the following observations made in patients with unilateral brain lesions could support the hypothesis of a causal link between the left lateralization of language and other brain asymmetries:(a)If different kinds of verbal and non-verbal cognitive activities can take advantage of the use of an implicit verbal mediation, then (at least some) non-verbal cognitive tasks should be impaired in aphasic patients with left-brain lesions. Furthermore, if the verbal LH is more involved in conscious and voluntary (and the non-verbal RH in more automatic) aspects of brain functions, then patients with left brain damage should be more impaired in intentional (and right brain-damaged patients in less conscious and more automatic) aspects of brain functioning.(b)If interferences should be avoided between the more controlled but slower language-mediated cognitive system and the more automatic and faster emotional activities (and spatial orienting of attention), then the latter should be best hosted by the right, rather than by the verbal left hemisphere.(c)If the competition for cortical space implies that areas homologous to those involved in language in the LH remained dedicated to other more basic sensory-motor functions in the RH, then patients with right brain damage should be more impaired not only on high-level perceptual tasks but also on non-verbal representational activities grounded on these sensorimotor activities.

## 3. Neuropsychological Findings That Could Allow to Check the Correspondence between Predictions Based on the Theoretical Impact of Language on Other Brain Functions and Data Obtained in Patients with Unilateral Brain Lesions

### 3.1. Data Obtained Studying the Relations between Verbal and Non-Verbal Disorders of Aphasic Patients

It has been explicitly acknowledged in Section 2.1.1. of this review that many different abilities are requested to solve non-verbal cognitive tasks and that a clear distinction must, therefore, be made between the non-verbal cognitive tasks that are significantly impaired and those that are less clearly impaired in aphasic patients. For this reason, it was considered appropriate to address this problem from the historical point of view, in order to give a more complete account of its many facets.

#### 3.1.1. Early Proponents of a Non-Verbal Cognitive Impairment in Aphasia

Several classical authors (e.g., Head [34] and Goldstein, [35]), often labeled as ‘noetic authors’, have given a general account of their investigations, showing that the cognitive impairment of aphasic patients (APs) extends beyond the purely linguistic domain. Head [34] considered this cognitive impairment as the consequence of a defect in the purposeful use of symbols and claimed that APs are not impaired on cognitive tasks that can be performed with simple perceptual activity, but are defective when an intermediate (verbal or non-verbal) symbolic activity is required by the task. Thus, the greatest cognitive impairment of APs should be observed on tasks that require an intermediate symbolic activity between the initial conception and the final execution of the act. Goldstein [34] also acknowledged that language influences thought formation because it is not only an advanced means of communication but can also play a critical role in developing and supporting thought activities. More specifically, Goldstein [35] claimed that the basic cognitive defect of APs consists of a “loss of the abstract attitude”, in which the subject detaches himself from the concrete immediate sensory components of a situation and relies on abstract rules and general concepts.

#### 3.1.2. The First Application of Methods Drawn from Experimental Psychology to the Study of Non-Verbal Cognitive Disorders in Aphasia

After these pioneers of the study of the cognitive impairment of APs, other researchers used designs and procedures drawn from the field of experimental psychology to investigate the non-verbal cognitive disorders of APs. The first author who used this methodology to test the assumptions of the ‘noetic’ authors was Bay [36,37]. This author showed that APs often perform poorly on conceptual modelling tasks because they fail to reproduce the features typical of the conceptual representation of different objects, but provided no control data on the modelling capacities of non-aphasic brain-damaged patients (BDPs).

More appropriate, from both the conceptual and the methodological point of view, was, a series of neuropsychological investigations conducted in the following years by De Renzi and coworkers [38,39,40,41,42,43,44,45,46,47,48,49]. These authors administered to aphasic and non-aphasic, left and right BDPs two partly separable subgroups of cognitive tasks: (a) the first consisted of ‘problem-solving’ tasks, such as the Raven’s Coloured Progressive Matrices (RCPM/[46,47,48]) and the Weigl’s Sorting test [39] that require patients to solve respectively a visual-spatial and a logical problem; (b) the second group consisted of ‘associative’ or ‘conceptual’ tests, such as the “Coloring Drawings of Objects” [41,45,47], the “Meaningful Sound Recognition” task [40,44,49] or the “Use of Objects” task [42], that require patients to match (using concrete pictorial material as stimuli) two different features belonging to the conceptual representation of the same object. (c) Furthermore, in some of these investigations, the specificity of results obtained on conceptual or problem-solving tasks was assessed by administering to the same patients high-level perceptual tasks, such as the Meaningless Sounds Discrimination [44,49], the ‘Ghent overlapping figures’ test [39,43], or the “Color Matching” test [41].

Results obtained in this complex series of investigations can be summarized as follows:Aphasic patients showed consistent impairment in both ‘problem-solving’ and conceptual tasks.Their specific cognitive defects seemed linked to the abilities tapped by the ‘conceptual associative’ tests [40,41,44], by Weigl’s test of abstract thinking [38] and by visual-spatial reasoning, investigated by the RCPM [46,47,48].On the contrary, right brain-damaged patients were selectively impaired on the high-level perceptual tasks, such as the Meaningless Sounds Discrimination [44,49], the “Color Matching” [41] or the ‘Ghent overlapping figures’ test [39,43].

In more recent years, results obtained by De Renzi and coworkers were substantially confirmed by different groups of research, who deepened the study of the conceptual disorders of APs or of their problem-solving defects. Furthermore, other lines of research supported the hypothesis of a greater involvement of the right hemisphere in more basic perceptual functions.

#### 3.1.3. The Relations between Non-Verbal Cognitive Impairment and Semantic-Lexical Disintegration in Aphasia

This subject was developed by Gainotti and coworkers, who assumed that only some components of aphasia may be intimately linked with the non-verbal cognitive impairment and that these components might be related to the comprehension and expression of concepts through language. These authors also surmised that a selective relation might exist between non-verbal cognitive impairment and disruption of the semantic-lexical level of language, whereas a more elementary language level (i.e., phonology) should be less relevant to non-verbal cognitive functions. This model was tested in a series of investigations, conducted in large groups of unselected aphasic and non-aphasic right and left BDPs and normal controls by means of the following tests: (1) comprehension of symbolic gestures [50]; (2) conceptual relationships between objects [51]; (3) drawing objects from memory [52]; (4) classificatory activities [53]; RCPM [54]. On all these tests APs scored significantly worse than non-aphasic (right and left) BDPs and this difference was particularly strong on the most complex tasks. For instance, in the study of classificatory activities, APs were only marginally impaired on a task of “class inclusion” that required them to put together pictures of ‘fruits’ or of ‘red items’. They were, on the contrary, much more impaired on a task of “class intersection” in which they were requested to select the item (e.g., the picture of ‘a cherry’) that was situated at the intersection between the above categories. Furthermore, in all these investigations a strong relationship was found between non-verbal cognitive impairment and the presence of a semantic-lexical disorder. Even if these results confirmed the existence in aphasia of an intimate link between—disruption of the semantic-lexical level of language and—impairment on non-verbal conceptual tasks, they did not prove a strict causal link between these two phenomena. In fact, a strong relation was also found between the presence of semantic-lexical disorders and poor results on the RCPM, which is a problem-solving test based on abstract material, rather than on stored conceptual representations. These results showed that only part of the non-verbal cognitive disorder of APs is due to a basic conceptual disorder, that expresses itself in both the verbal and the non-verbal modality, whereas another part could be due to the inability to solve the task through an intermediate propositional use of covert language, applying- the rules of grammar to organize the lexicon in novel and creative ways.

#### 3.1.4. The ‘Defective Isolation of Individual Features of Concepts’ in Aphasia

A more theory-driven line of research about the specific features of conceptual disorders shown by APs was followed by Cohen and associates. These authors reasoned that one of the main cognitive functions of language consists in analyzing external stimuli focusing attention on their specific features and proposed that non-verbal cognitive defects of APs might be due to a ‘defective analytical isolation of individual features of objects or concepts. Support to this hypothesis came from results obtained on non-verbal matching tasks in which subjects had to decide which of two pictures was more closely related to a third target picture [55,56,57,58]. In one condition the decision had to be based on the existence of a concrete common situational context, whereas in the other condition it had to be based on the isolation and abstraction of critical features of the depicted objects. Aphasic patients did not score worse than control subjects on the first type of task but scored significantly worse than any other control group on the second task, which required comparing different concepts with respect to individual features.

#### 3.1.5. Impaired Reasoning and Non-Verbal Problem-Solving in Aphasia

Baldo and coworkers [59,60,61] developed a research line assuming that, since human language serves to bolster our capacity for reasoning and problem-solving, APs should be impaired on non-verbal complex problem-solving tasks. In a first study [59] they administered to aphasic and non-aphasic stroke patients the Wisconsin Card Sorting Test (WCTS), which requires sorting cards into different piles, based on three different salient criteria and the RCPM. They showed that in their stroke patients there was a significant correlation between language measures, such as comprehension and naming, and performance on the WCST and RCPM, suggesting that language plays a role in complex problem solving, possibly through covert language processes. In a second investigation, Baldo et al. [60] administered the RCPM, to a larger group of left hemisphere stroke patients, making a separate analysis of items in which the solution could be determined by a simple visual pattern-matching, and of items whose solution required a more complex relational reasoning. These authors also used a voxel-based lesion-symptom mapping (VLSM) procedure to relate patients’ performance on these different items of the RCPM with areas of brain damage. Performance on the relational-reasoning items was disproportionately affected in aphasic patients and deficits on the relational reasoning problems were associated with lesions in the left middle and superior temporal gyri, essential for language processing. On the contrary, results on the visual pattern-matching condition were associated with lesions in temporo-occipital areas subserving visual processing. In a further paper, Baldo et al. [61] presented new data from a large group of APs that showed a dissociation in performance between two non-verbal tasks of the Wechsler Adult Intelligence Scale (Picture Completion vs. Picture Arrangement tasks) that require differing degrees of reasoning. The ‘Picture Completion’ task can, indeed, be solved with a simple perceptual activity, whereas in the ‘Picture Arrangement’ task the subject must detect the differences between pictures that allow to arrange them to form a reasonable and meaningful story. In agreement with Head’s [34] claim that APs are not impaired on cognitive tasks that can be performed with simple perceptual activity, but are defective when an intermediate verbal activity is required by the task, Baldo et al. [61] showed that language impairment leads to worse performance on the task that places a greater demand on reasoning than on perceptual abilities, even if both tasks do not require an overt language production.

#### 3.1.6. The Hypothesis Assuming That Words Affect Ongoing Cognitive and Perceptual Processes via Top-Down Feedback

The hypothesis—assuming that language does not simply allow us to communicate about our experiences, but also transforms cognition and perception, allowing humans to access and manipulate mental representations in novel ways—was recently developed by Lupyan and coworkers [16,62,63].

Lupyan [16] reviewed a growing number of studies showing that language, and specifically the practice of labeling, can exert rapid and pervasive effects on putatively non-verbal processes such as categorization and visual discrimination. Focusing attention on the putative effects of language on color categorization and color perception, he proposed a ‘label-feedback hypothesis’ which assumes that language may produce a transient modulation of ongoing perceptual (and higher-level) processing. In the case of color, this means that after learning that certain colors are called “green,” viewing a green object becomes a hybrid visuo-linguistic experience, by which our everyday experiences of seeing become affected by the verbal term, which in turn makes the visual representation more categorical. Lupyan and Mirman [62] systematically tested the link between language defect and categorization impairment by comparing categorization performance in APs and in education-matched normal controls on tasks in which the categorization criterion was either “high-dimensional” (i.e., the objects shared many features, such as “farm animals”) or “low-dimensional” (i.e., the objects shared one or a few features, such as “things that are green”). Aphasic patients were selectively impaired on low-dimensional categorization and their selective impairment was correlated with the severity of their naming impairment, indicating, in agreement with the positions of Cohen et al. [55,56,57,58] that language impairment impacts categorization specifically when that it requires focusing attention and isolating individual features of concepts. Consistent with the same hypothesis were also results of a study in which Pauly et al. [64] tested left or right hemisphere stroke patients on a speeded color discrimination task in which two factors were manipulated: (1) the categorical relationship between the target and the distracters and (2) the visual field in which the target was presented. Similar to controls, the RH patients were faster in detecting targets in the right visual field when the target and distracters had different color names compared to when their names were the same. This effect was absent in the LH patients, suggesting that injury to the left hemisphere handicaps the automatic activation of lexical codes.

Even more recently, Lupyan et al. [63] reviewed the behavioral and electrophysiological evidence for the influence of language on perception, with an emphasis on the visual modality. They showed that the effects of language on perception can be observed both in higher-level processes such as recognition and in lower-level processes such as discrimination and detection. In any case, a consistent finding of these investigations was that language causes us to perceive in a more categorical way.

#### 3.1.7. Overall Evaluation of Results Obtained Studying the Relations between Verbal and Non-Verbal Disorders of Aphasic Patients

Taken together, results reviewed in the different parts of this section clearly show that two main difficulties are usually shown by APs in front of non-verbal cognitive tasks. The first concerns the semantic-lexical level and consists of a difficulty to construct appropriate conceptual categories and to detect the typical features of individual objects or concepts. The second defect involves the syntax at the sentence level and concerns the capacity of reasoning and solving complex visual-spatial or logical problems through an intermediate propositional use of covert language.

It must, however, be remembered that, besides documenting the impairment of APs on non-verbal conceptual tasks, some studies conducted in aphasic and non-aphasic BDPs had also shown an opposite dissociation between right and left BDPs on non-verbal perceptual tasks (e.g., [40,41,44,49]). Aphasic patients obtained, in fact, worse results on conceptual tasks, whereas patients with RH damage performed significantly worse on purely perceptual tasks. These results, which suggest that language-mediated cognitive abilities are more represented in the left LH, whereas simple perceptual functions are more represented in the RH will be discussed in Section 3.4. of this review dedicated to the competition for cortical space determined by the development of language within the left hemisphere.

### 3.2. Data Concerning More General Aspects of Thought (i.e., the Levels of Consciousness and of Intentionality) That Could Be Influenced by the Left Lateralization of Language

In Section 2.1 of this review, I have reminded Jackendoff’s [14] statement that language can influence not only the content but also the structure of thought and I have assumed that the more general aspects of thought that could be shaped by language may mainly concern the levels of consciousness and of intentionality. In the present section I intend to develop these general assumptions, reviewing the clinical data which suggest: (a) that the RH functioning may be considered as more automatic and unconscious than the LH functioning and (b) that this difference between the ‘verbal’ and the ‘non-verbal’ hemisphere in terms of consciousness and intentionality may increase when we pass from the more cognitive functions, typical of the left hemisphere, to the emotional and visual-spatial functions, typical of the right hemisphere.

#### 3.2.1. The Distinction between Propositional and Automatic Speech in Aphasia and the Role of the Right Hemisphere

The distinction between ‘propositional’(voluntary) and ‘non-propositional’(automatic) speech has been used since Jackson’s [65,66] classical papers on this subject, to understand different aspects of the aphasic speech and language disorder. This distinction was clear in Broca’s [1] description of his famous patient Leborgne, who was speechless, but often said “tan” … and “Sacré nom de Dieu!”. Drawing on this distinction, Head [34] stated that non-propositional speech appears on the foreground in both receptive and expressive aphasia and Marie [67] noted that a dissociation between voluntary and automatic speech is a common clinical observation in aphasia. In more recent years, attempts have been made to operationally define non-propositional language and several authors have reported patients, who, in a context of global aphasia, showed an advantage of automatic over intentional (propositional) speech. As for the first issue, Code [68] claimed that non-propositional language is characterized by the automatic production of overlearned sequences, such as counting, reciting the days of the week or the months of the year, reproducing some conventional greetings, repeating or completing familiar phrases As for the second issue, Lum and Ellis [69] gave to a group of aphasic patients three pairs of these tasks that compared the production of the same items in either propositional or automatic contexts. All patients showed an automatic advantage on at least one pair of tasks, and there were no examples of better performance on the propositional than on the automatic version of any task. Analogously, Snowden and Neary [70] reported a patient, with progressive classical anomia, resulting from focal degeneration of the left hemisphere, who demonstrated strikingly preserved naming performance on automatic compared to nominative tasks. This substitution of propositional with automatic speech was explained by many authors by assuming that, when the LH is damaged, the right mediates automatic speech production. This suggestion had been first introduced by Jackson [66] with this expression: “The right hemisphere is the one for the most automatic use of words, and the left the one in which automatic use of words merges into voluntary use of words—into speech”. A role of the RH in residual speech was also postulated on the basis of symptomatic worsening in left aphasic patients after temporary RH inactivation by intracarotid amobarbital injection [71] or diminution of residual speech brought about by a new stroke to the previously intact right hemisphere [72,73]. Furthermore, Ryding et al. [74], who examined with the regional cerebral blood flow technique 15 non-aphasic right-handed subjects with some neurological symptoms on reciting the days of the week and humming a nursery rhyme with a closed mouth, observed significantly more activity in the right than in the left hemisphere during automatic speech. A role of the RH in automatic speech production was also suggested by Speedie et al. [75], Graves and Landis [76] and Van Lanker and Cummings [77]. The first authors reported the case of a bilingual patient whose automatic speech was disrupted following a hemorrhage involving the right basal ganglia. The patient was not aphasic but was unable to produce certain types of automatic sequences such as counting, reciting prayers, singing familiar songs, in both his first and in his second language. Graves and Landis [76] compared the production of automatic and propositional speech in aphasic speakers and suggested that automatic speech was produced by the RH. This suggestion was due to the fact that, by measuring mouth openings during the production of automatic and propositional utterances, they found that opening of the right side of the mouth was greater for spontaneous speech, repetition, and word list generation, while the opening of the left side of the mouth was greater for serial speech and singing. Finally, Van Lanker and Cummings [77] described a right-handed adult who, shortly after surgical removal of his LH for treatment of a tumor, was profoundly aphasic and unable to produce propositional responses to questions but could produce automatic forms of speech.

#### 3.2.2. The Selective Disruption of the Voluntary Execution of Movements in Left Brain-Damaged Patients with Apraxia

According to recent definitions of the term apraxia [78,79], this defect covers a wide spectrum of disorders, that have in common an inability to perform a skilled or learned act that cannot be explained by an elementary motor or sensory deficit or by a language comprehension disorder. The term apraxia was first used by Steinthal [80], who described a faulty and awkward tool used by an aphasic patient and concluded that “apraxia is an obvious amplification of aphasia”. Some years later Jackson [66] documented the distinction between automatic and voluntary execution of movements in apraxia and this distinction was afterwards systematically confirmed in bucco-facial apraxia and in ideo-motor apraxia. Since the first description of this disorder, research and clinical experience have confirmed the regular association of apraxia with aphasia but the interpretation of this association has been controversial, making it difficult to understand the relations between the voluntary/automatic dissociations observed in aphasia and in apraxia. Some authors have stressed the similarity between language and gesture production drawing on theoretical or empirical reasons. Thus, Roby-Brami et al. [81] claimed that the neural systems supporting these functions are predominantly located in the LH, drawing on the hypothesis that postulates strong evolutionary links between language and praxis, including the possibility that language was originally gestural. Willems and Hagoort [82] also concluded that there is strong evidence on the interaction between speech and gestures in the brain starting from an overview of studies in cognitive neuroscience that examined the neural underpinnings of links between language and action. On the other hand, Goldenberg and Randerath [83] noted that neither aphasia nor apraxia are indivisible entities because both diagnoses embrace diverse manifestations that may occur more or less independently from each other. Thus, the question of whether apraxia is always accompanied by aphasia may lead to conflicting answers depending on which of their manifestations are considered.

In a recent in-depth discussion of the lateralization of mechanisms for language and praxis production, Kroliczak et al. [84] have suggested that the developmental order of handedness, praxis, and speech acquisition could be a critical factor to clarify this issue. In fact, if hand-preference mechanisms get implemented in the brain considerably earlier than speech production mechanisms (as according to Johnston et al. [85], could be quite likely in lefthanders), then the factors underlying unimanual manipulation and hand-preference mechanisms could jointly exert a pressure sufficient for the segregation of the (right-lateralized) praxis from the left-lateralized language mechanisms. If, on the contrary, speech development precedes, or is at least simultaneous with the acquisition of hand preference, then praxis and language could/should be tightly linked, by sharing common mechanisms or more general processing resources (Kroliczak, et al. [86]). Coming back from these general problems to the selective disruption of the voluntary execution of movements in patients with apraxia, it must be said that, as in the study of the distinction between propositional and automatic speech in aphasia, some authors have explicitly assumed that RH structures may underpin the automatic execution of movements that cannot be produced intentionally. Thus, Rapcsak et al. [87] reported the observation of a patient affected by a massive stroke that had resulted in the virtually complete destruction of the left hemisphere. This patient was severely impaired in pantomiming transitive gestures with the left hand and in reproducing novel non-symbolic hand and arm movement sequences but could produce overlearned habitual actions such as actual object use and intransitive gestures. Based on these findings, Rapcsak et al. [87] proposed that the praxis system of the RH may be strongly biased toward “concrete” or context-dependent execution of familiar, well-established action routines. It could be, however, critically dependent on transcallosal contribution from the LH for control of the left hand in “abstract” or context-independent performances of transitive movements and in learning novel movement sequences. An action model consistent with this hypothesis has been advanced by Tucker and Ellis [88], who noted that the concept of affordance, originally proposed by Gibson [89], has been incorporated in recent models of interactive behavior and proposed that visuo-motor relations between objects and actions can “automatically” activate a motor response for object-use, even if this response is not required by the task. The praxis system of the RH could, therefore, be automatically primed by common objects or concrete situations, without the need for high-level object recognition or action programming processes, required by the context-independent praxis system of the left hemisphere.

### 3.3. The Disruption of Automatic Components of Behaviour in Patients with Right Hemispheric Lesions

Investigations reviewed in the last two sections of this paper have clearly shown that lesions of the LH selectively disrupt the most voluntary/propositional aspects of language and gesture, leaving intact their automatic components. Some of these investigations have also suggested that the RH could be responsible for the production of the automatic components of speech and gesture, but evidence supporting this suggestion was not very strong, since it mainly consisted of anecdotical observations. Much more consistent evidence supporting the assumption of a greater involvement of the RH in automatic components of behavior comes from the study of disorders resulting from RH damage. Particular attention will be paid in this review to the greater involvement of the RH in emotional functions, to the mechanisms underlying the spatial orienting of attention in unilateral spatial neglect, and to the loss of face familiarity feelings in acquired prosopagnosia.

#### 3.3.1. The Greater Involvement of the Right Hemisphere in Emotional Functions

It has been reminded in Section 2.2.1 of this review that, according to many theorists (e.g., [28,29,30,31,90,91,92,93]), the emotional and the cognitive systems are the two basic adaptive systems that allow the organism to face a partially unpredictable environment. Even if both these adaptive systems are phylogenetically advanced and are based on the integrated activity of different components, their aims are different. The emotional system is considered as a more primitive emergency system, whereas the cognitive system is viewed as a more complex and evolved system. Therefore even if both these systems are based on the integrated activity of components that must process ongoing information, select the most appropriate response and keep a record of the whole event, the manner in which each system accomplishes these tasks is different. Thus, the ‘emotional’ analysis of sensory information is global, rapid, and unconscious, aiming simply to evaluate if the external situation is pleasant or dangerous, whereas the ‘cognitive’ analysis is objective and exhaustive but slower and often leads to gathering further information. Analogously, the emotional system automatically selects the most suitable response from a small number of innate schemata, corresponding to the survival needs of the species, whereas the responses selected by the cognitive system are complex and can contrast with the quick but inappropriate responses triggered by the emotional system. For these reasons, a strong overlap between the structures underlying the more automatic functioning of the emotional system and the more controlled functions of cognitive systems could raise more problems than an interaction between spatially separated, although strongly interconnected, brain networks. From the historical point of view, right lateralization of emotions was first proposed by Luys [94] who suggested the existence in the RH of an “emotion” center, complementary to the “intellectual” centers in the LH. A similar suggestion was afterwards advanced by Gainotti [95], who described a different (‘catastrophic’ vs. ‘indifferent’) emotional reaction to the disability of the right and left BDPs. This author considered the “catastrophic reaction” of (aphasic) left BDPs as a dramatic, but psychologically appropriate form of response to a catastrophic event but regarded the indifference reaction’ of patients with severe RH damage, as an abnormal reaction to a dramatic event. To explain the contrast between the ‘indifference’ and the ‘catastrophic’ reactions, Gainotti [95] suggested that the RH may be dominant for emotion, just as the LH is dominant for language, and argued that the emotional reaction may be inappropriate in right BDPs because structures disrupted in these hemispheres are crucially involved in emotional functions. I do not intend to review here the clinical and experimental data which have confirmed the prevalent role of the RH in different components of emotions. I will only remind that a greater involvement of the RH in emotional functions has been documented by data gathered along lines of research concerning: (a) a greater involvement of the RH in the communicative aspects of emotions (see [96,97,98,99,100] for reviews); (b) a greater role of the right amygdala in the quick automatic evaluation of emotional stimuli ([101,102]; see [103,104] for reviews); (c) the preferential involvement of the right ventro-medial prefrontal cortex in the integration between cognition and emotion and in the control of impulsive reactions (e.g., [105,106,107] see [108] for review); (d) the selective contribution of the right insula to the conscious experience of emotion (e.g., [109,110,111,112]; see [108] for review).

Within these investigations, I would, however, dwell briefly on data that have shown a greater role of the right amygdala in the quick automatic evaluation of emotional stimuli, because these data are relevant to the problem of the relations between language and consciousness that has been introduced in Section 2.1.2 of this review.

This line of research was triggered by two papers by Morris et al. [101,102], who investigated the mechanism of an unconscious form of emotional learning in which an aversively conditioned masked emotional face elicited an unconscious emotional response. In a first PET study [101], these authors showed that the masked presentation of a conditioned emotional face provoked a significant neural response in the right, but not the left amygdala, whereas the unmasked presentation of the same stimulus enhanced neural activity in the left but not the right amygdala. In their second paper, Morris et al. [102] tried to clarify the mechanism by which this unconscious form of emotional learning could be accomplished, focusing on the distinction between a cortical and a subcortical route [30,113] through which perceptual stimuli might reach the amygdala. They found that, during the unconscious (masked) presentation of conditioned emotional stimuli, an increased correlation was observed between activity in the right amygdala, pulvinar, and superior colliculus, whereas no masking-dependent changes in correlation were observed among the same subcortical structures and the left amygdala. Morris et al. [102] concluded that emotionally laden stimuli can be processed without conscious awareness by the RH subcortical pathway, mediating unconscious emotional learning whereas the left amygdala could play a role in the more complex and conscious process of stimulus evaluation. These conclusions are consistent with the results of a meta-analysis of fMRI studies of amygdala responsivity to emotional stimuli [114], which confirmed an LH lateralization for (conscious) verbal stimuli and an RH lateralization for (masked) emotionally laden stimuli. Results congruent with these conclusions were also obtained in patients with brain lesions by Ladavas et al. [115] and by Glascher and Adolphs [116]. The first authors studied the cognitive and physiological emotional responses of a split-brain patient to subliminal and above threshold presentation of emotional and non-emotional stimuli and showed that only the RH is able to produce an appropriate autonomic response to the presentation of emotional material, in the absence of a conscious evaluation of the eliciting stimulus. Glascher and Adolphs [116] presented both subliminal and supraliminal emotional stimuli to patients with left, right, and bilateral amygdala damage and observed impaired skin conductance responses only after right amygdala damage. All these data confirm the difference between the automatic and unconscious processing of emotional information made by the non-verbal right hemisphere and the controlled and conscious processing of the same information made by the verbal left hemisphere.

#### 3.3.2. Lack of Disease Awareness (Anosognosia) and the Right Hemisphere Dominance for Emotions

Another condition that could be linked to the unconscious processing of emotionally laden conditions by the RH is the lack of awareness (anosognosia) of hemiplegia first described by Babinski [117] in patients with RH stroke. Babinski [117] argued that the RH might have a special role in the pathophysiology of disease unawareness and subsequent anatomo-clinical investigations confirmed this suggestion, even if the question was complicated by the confounding effects linked to the presence of aphasia in patients with LH lesions and of unilateral spatial neglect in anosognosic patients with RH stroke (see [118] for review). Furthermore, since anosognosia for motor impairment is only an instance of a complex and widespread disturbance of disease awareness, which can also concern various motor, sensory and cognitive disorders, Gainotti [119] tried to determine whether the unawareness phenomena are linked to the RH dominance for emotions in patients with degenerative brain disease, where a prevalence of right-sided lesions is often associated with emotional and behavioral disturbances. Results of the review confirmed that the neural correlates of anosognosia are often right lateralized in patients with degenerative brain diseases and that the role of right frontal lesions is much greater when the loss of insight concerns emotion-linked aspects of personality or behavior than when it concerns particular aspects of cognition or memory. Taken together, results of investigations concerning hemispheric asymmetries in emotional functions not only confirm a right lateralization of emotions but also show that the processing of emotional stimuli is much less conscious (and the awareness of emotionally laden conditions more defective) in the right than in the left hemisphere. It might be objected that the evidence that patients with RH stroke sometimes showed anosognosia for hemiplegia suggests that RH contributes to conscious processing of our body and this has been called the “self-awareness-anosognosia” paradox [120,121]. According to Morin [120], this paradox could be due to the fact that self-awareness and anosognosia do not constitute unitary concepts and that impaired awareness of deficit is mostly caused by problems with self-monitoring and pre-/post-brain damage comparisons of performance. On the other hand, Daini [121] has reworked Gazzaniga’s idea [122,123] of an “Interpreter”, located in the left hemisphere, suggesting that anosognosia of left-sided hemiplegia may be due to a disconnection of structures processing information about the paralyzed limbs from the left Interpreter. This model, therefore, assumes that the left hemisphere may be relevant for self-consciousness and that the right hemisphere damage does not affect the areas strictly involved in self-consciousness but instead induces a “disconnection” between what is processed in the right hemisphere and the self-consciousness-related system in the left hemisphere.

#### 3.3.3. The Loss of Automating Orienting of Attention toward the Left Side of Space in Unilateral Spatial Neglect

Unilateral Spatial Neglect (USN) is the most frequent and dramatic behavioral defect of patients affected by RH damage. This disorder was defined by Heilman et al. [124] as “a failure to report, respond or orient to novel meaningful stimuli presented to the side opposite to a brain lesion”. The success of this definition was due to the fact that it described elegantly and simply the main features of USN, but its limit consisted in a failure to consider the heterogeneity of the defect in the spatial orienting of attention, typical of USN. Since the pioneering work of James [125], who distinguished between a “passive, reflex, non-voluntary, effortless” and an “active and voluntary” form of attention, it is, indeed, acknowledged that attention can be directed to an object in space either in a reflexive or in a more controlled way. If we take into account this fundamental distinction, we must acknowledge that Heilman et al.’s [124] definition failed to take into account two important aspects of the unilateral neglect syndrome. The first is that the ‘endogenous’ (conscious and controlled) forms of contralateral orienting of attention are relatively spared in these patients [126,127], whereas the ‘exogenous’ (automatic) forms are much more impaired (see [128,129,130] for reviews). The second is that USN occurs not only because stimuli arising in the contralateral half space are unable to attract the patients’ attention, but also because attention is automatically captured by stimuli lying in the ipsilateral half space. Mark et al. [131] have given an elegant demonstration of this phenomenon, by administering to neglect patients two equivalent versions of a cancelation task. In the experimental (‘erasing’) condition, patients canceled lines by erasing them, whereas, in the control (‘drawing over’) condition, they simply drew a line over them. Since in any case patients began their activity from the right part of the sheet, in the experimental condition, stimuli disappeared from that part of space, whereas in the control condition they persisted on the right half of the sheet. There were significantly more left-sided omissions in the ‘drawing over’ than in the ‘erasing’ condition, showing that the severity of neglect is increased by the presence of stimuli on the right half space, that automatically attracted the patient’s attention. These results have been confirmed by several other authors (e.g., [132,133,134,135]) who have shown that the presence of irrelevant stimuli on the right half space automatically captures the attention of neglect patients, increasing the severity of this syndrome. As for the endogenous mechanisms of spatial orienting of attention, Gainotti [128] noted that traditional forms of neglect rehabilitation are usually based on a conscious, verbally induced exploration of the neglected half space (e.g., [136]) and could thus be preferentially mediated by the left hemisphere. Consistent with these neuropsychological findings are the models concerning the neuroanatomical bases of neglect and the brain mechanisms underlying exogenous and endogenous forms of the spatial orienting of attention. [137,138,139]. Models of attentional orienting in humans have, in fact, proposed that a ‘dorsal’ fronto-parietal network might be involved in the endogenous orientation of spatial attention, whereas a more ‘ventral’ network could underlie the exogenous forms, detecting unexpected but behaviorally relevant events. The dorsal network could be symmetrically represented in both hemispheres, whereas the ventral network should be more active in the RH than in the language-dominant left hemisphere [140,141].

#### 3.3.4. The Loss of Face Familiarity Feelings in Acquired Prosopagnosia

Acquired prosopagnosia is a disorder of visual recognition specific to faces, associated with bilateral occipital or temporal lesions but with a strong prevalence of RH damage when lesions are unilateral. In an influential review of ‘face recognition’ processes, Gross and Sergent [142] explicitly claimed that: “all prosopagnosics display the same functional impairment—an inability to experience a feeling of familiarity at the view of faces of known persons and to identify these faces”. As in the case of prosopagnosia, a relation between the RH and face familiarity feelings has been repeatedly demonstrated in healthy subjects, by asking them to make familiarity judgments about faces presented separately to the right and left visual fields (e.g., [143]) or by studying the lateralization of event-related potentials evoked by face familiarity [144]. In keeping with these results, Gainotti and Marra [145] documented a selective defect of face familiarity feelings in patients with unilateral lesions of the anterior or the posterior parts of the right temporal lobes, who showed a defect of familiar people recognition. Similar data were obtained by Borghesani et al. [146] studying, in a large sample of patients with neurodegenerative diseases, the neuroanatomical substrates of three different steps of famous face processing. These authors correlated, using voxel-based morphology, whole-brain grey matter volumes with scores on three experimental tasks that targeted respectively: (a) face familiarity judgment, (b) retrieval of semantic/biographical information, and (c) face naming. They showed that, although performance in naming and semantic information retrieval correlated significantly with grey matter volume in the left anterior temporal lobe, familiarity judgment correlated with integrity of the right anterior middle temporal gyrus. The automatic nature of familiarity feelings generated by the view of a known face is documented by the subjective common experience that we feel in these situations and by general dual-process theories (e.g., [147,148,149]) that acknowledge that recollection reflects an analytic, conscious, controlled process, whereas familiarity is a rather automatic process. This position is also consistent with results obtained by Palermo and Rhodes [150] in a review of the general problem of the attentional mechanisms involved in recognition of facial identity and facial expression and by Jung et al. [151] and Yan et al. [152] in more recent studies dealing with the automaticity of familiar-face identification. Palermo and Rhodes [150] substantially confirmed that faces, being salient emotional stimuli that convey crucial information for social interactions, are at least in part automatic. Jung et al. [151] showed that automatic face identification is dependent on prior familiarity and Yan et al. [152] showed that familiarity enhances the automatic processing of identity judgment. All these data support the general hypothesis which assumes that automatic and controlled levels of processing may have a different representation in the right and left hemisphere, the right being more involved in the earliest automatic and the left in the more controlled and propositional levels [97,153] of various cognitive and emotional functions.

### 3.4. The Greater Representation of Perceptual Functions in the Right Hemisphere and the Implications of this Asymmetry for Representational Activities

In Section 2.3. of this review it had been assumed that, due to competition for cortical space, areas homologous to those involved in language processing in the LH could remain dedicated to more basic perceptual functions in the right side of the brain. This assumption has been supported by results obtained by De Renzi et al. [39,41,43,44,49] and reported in Section 3.1.2. of this review, which has documented a selective impairment of right BDPs on purely perceptual high-level visual and auditory tasks. Besides the results of these studies, data consistent with a greater contribution of the RH to perceptual functions have been obtained by other investigations conducted with different methodologies in healthy subjects or in BDPs. In particular, this hypothesis has been confirmed by studies conducted in patients with unilateral brain lesions by Grossman and Wilson [154] in the visual modality and by Schneider et al. [155] in the auditory modality. Grossman and Wilson [154] assessed visual discrimination and categorization of pictures of living and non-living items in patients with unilateral brain lesions and found that anomalies in categorizing, but not in discriminating the fruit and vegetable pictures prevailed in left BDPs, while the reverse was true for the right BDPs. Schneider et al. [155] showed that right BDPs failed to discriminate between acoustically related non-verbal environmental sounds, whereas patients with LH lesions tended to confuse semantically related sounds. Many other investigations aiming to study hemispheric asymmetries on visual perceptual tasks in normal subjects have used the tachistoscopic lateralized presentation of visual stimuli under masked or unmasked conditions. (e.g., [156,157,158]) or a manual reaction time identification task to laterally presented stimuli (e.g., [159,160]). All these investigations have shown that nonverbal visual stimuli produce faster and more accurate responses for right- than for left-hemisphere presentations. Similar methodologies have been used to assess hemispheric lateralization of visual perception in patients with temporal lobe epilepsy [161] and in split-brain patients [162]. In this last study, subjects were given four simple visual-matching tasks, in which two stimuli were presented briefly to one visual hemifield and the patient was asked to discriminate whether they were the same or different. The first three tasks (orientation discrimination, vernier offset discrimination, and size discrimination) were all spatial in nature and were performed better by the right hemisphere. Hougaard et al. [163] also confirmed the greater RH representation of visual functions with a functional neuroimaging study in which authors investigated the asymmetry of fMRI-BOLD responses to a simple checkerboard visual stimulation. Hougaard et al. [163] found a right lateralization of early visual cortical areas and higher-level visual processing, involved in visuospatial attention and showed that this right lateralization was partly explained by an increased grey matter volume of the early visual areas in the right hemisphere. Few other investigations (e.g., [164,165,166]) have studied in healthy subjects hemispheric asymmetries on auditory perceptual tasks using the dichotic listening procedure. These studies have consistently, shown a left-ear superiority for musical or environmental sounds. All things considered, it seems, therefore, possible to conclude that, besides the strong left lateralization of language, there is also a greater representation of perceptual functions in the right hemisphere.

#### 3.4.1. Implications of the Greater Contribution of the Right Hemisphere to the Visual and Auditory Perception for Representation of Non-Verbal Conceptual Knowledge

A central tenet of the Embodied Cognition Framework, [167,168,169] is that semantic knowledge and other representational activities are grounded in sensorimotor activities. These are automatically engaged during online representational processing, re-enacting modality-specific patterns of activity, normally evoked during perception and action. Barsalou [167] also acknowledged that in humans, this re-enactment of the sensorimotor systems is closely integrated with the linguistic system and proposed that humans’ powerful symbolic capabilities emerge from interactions between language and simulation. On the basis of these premises, it seems logical to assume that the strong prevalence of language in the LH and the greater RH involvement in sensorimotor functions should lead to the development within the left and right hemispheres of two different ‘higher-order convergence zones’ [170,171], supporting respectively verbal and non-verbal forms of conceptual knowledge. Results consistent with this hypothesis were obtained by Snowden et al. [172] and Gainotti [173], who studied the patterns of famous people identification through non-verbal (face and voice) and verbal (name) modalities in early Semantic Dementia (SD) patients. In these stages of the disease, it is, indeed, possible to observe a prevalent degeneration of the left and right anterior temporal lobes (ATLs), which are critically involved in semantic representations. Snowden et al. [172] found that subjects with a predominance of left ATL atrophy were more impaired in person identification through names than through faces, whereas patients with more severe right temporal degeneration presented the opposite pattern. Furthermore, these authors showed that better identification of famous faces was associated with superior performance on the picture, compared with the word version of the Pyramids and Palm Trees (PPT) test of conceptual thinking [174], suggesting that objects and famous people might be represented in the same (visual) format in the right temporal lobe. Gainotti [173] confirmed that different patterns of impaired recognition of familiar people can be observed in patients with right and left ATL pathology because the loss of personal semantics was greater from faces and voices when atrophy mainly affected the right ATL and from names when the left ATL was preferentially damaged. These data have been more recently confirmed by results of an important study in which Woollams and Patterson [175] evaluated the cognitive consequences of the left-right asymmetry of atrophy in semantic dementia. In a large-scale case-series study, these authors found that patients with predominantly left atrophy obtained significantly lower scores in picture naming, whereas patients with right ATL atrophy were significantly more impaired on the picture version of the PPT test. The hypothesis assuming that the left and right hemispheres may underlie verbal and non-verbal forms of semantic knowledge was also confirmed by results of functional imaging studies (e.g., [176,177,178]) that addressed the question of material-specificity in conceptual and person-specific semantic knowledge. Thierry et al. [176] compared semantic processing of spoken words to the equivalent processing of environmental sounds and showed that words enhance activation in left superior temporal regions, while environmental sounds enhance activation in a right posterior superior temporal region. Thierry and Price [177] compared conceptual processing of verbal and non-verbal stimuli in both visual and auditory modalities and found that left temporal regions were more involved in comprehending words (heard or read), whereas the right temporal cortex was more involved in the comprehension of environmental sounds and images. Hocking and Price [178] presented simultaneously to their subjects one visual (written object name or picture) and one auditory (spoken object name or object sound) stimulus and instructed them to decide whether these stimuli referred to the same object or not. Verbal matching increased activation in a region of the left superior temporal sulcus involved in phonological processing, whereas non-verbal matching increased activation in a right fusiform region involved in structural and conceptual object processing (see [179,180] for reviews). The model assuming that the RH may mainly support non-verbal forms of semantic knowledge was also confirmed by results of a study in which Sunquist et al. [181] investigated the connections of the right and left ATLs with the perceptual information implemented by posterior cortices. This study, which combined semantic assessments of verbal and nonverbal stimuli and MRI-based fiber tracking, showed that the volume of tracts connecting the right ATL with the posterior–superior temporal cortices correlated with scores of nonverbal semantics, suggesting an integration between perceptual representations of visual objects in the right posterior mid/superior temporal cortex and their conceptual non-verbal representations within the right ATL.

#### 3.4.2. Other Representational Activities, Based on Perceptual Functions, That Are Preferentially Linked to the Right Hemisphere

The greater involvement of the RH in visual-perceptual functions, besides explaining the leading role played by this hemisphere in non-verbal conceptual knowledge, could also elucidate the prevalence of this hemisphere in other representational activities, that overlap in part with those concerning the unilateral spatial neglect, reported in Section 3.3.3. because they mainly concern the leading role of the right hemisphere in visual-spatial activities. We have seen, in the Introductory section of this review that already Kosslyn [8] had stressed the complementary role plaid by the left hemisphere in language functions and by the right hemisphere in language and in visual-spatial activities, as this author had suggested that spatial relations could be dichotomized into abstract (categorical) relations, related to the left hemisphere, and more concrete (coordinate) spatial relations related to the right hemisphere. A similar dichotomy had been incorporated by Baddley [182] in his model of working memory because this author had maintained that working memory disorders should be different in patients with unilateral right and left-brain lesions, concerning mainly the phonological loop domain in patients with left-sided lesions and the visuospatial sketchpad domain in these with right-sided damage. The link between RH lesions and disruption of spatial activities has been repeatedly confirmed by empirical studies (e.g., [183,184,185] and by investigations specifically designed to check the Kosselin’s and Baddley’s models (e.g., [186,187]).

De Renzi [183] has, indeed, shown that, within the visual representational activities, disorders of space exploration and cognition are more frequently met in patients with right than with left-brain damage. Heide et al. [184] found an asymmetry between left and right posterior parietal (PPC) lesions in remapping visual representations in conjunction with eye movements, which matches the clinical consequences of lesions of the temporo-parietal junction in humans. Pisella et al. [185] reviewed evidence showing an RH dominance for visuo-spatial processing and representation in humans, showing that visual disorganization symptoms are observed in typical RH defects, such as neglect and constructional apraxia. All these findings are consistent with data previously reported in Section 3.4. of this review, namely with results obtained in split BDPs by Corballis et al. [162] and with greater RH representation of visual functions documented by Hougaard et al. [163] with a functional neuroimaging study of a simple checkerboard visual stimulation. On the other hand, Kosselin’s and Baddley’s models were supported by Van der haam et al. [186] and by Wagner et al. [187]. Van der haam et al. [186] could confirm Kosselin’s model showing that LH patients presented a specific defect in processing categorical stimuli whereas RH patients were impaired in all conditions except for the categorical condition. On the other hand, Wagner et al. [187] investigated material-specific lateralization of working memory in patients with unilateral right or left temporal lobe epilepsy (TLE) and showed that patients with right TLE were more impaired on the visuospatial working memory task, whereas patients with left TLE showed more intrusion errors on the verbal working memory task

Another visual representational defect that is more frequently observed in patients with posterior RH lesions is prosopagnosia, a selective defect of face recognition that can be associated with damage confined to the RH [145,188,189] and which has been shortly discussed in Section 3.3.4 of this review

If we shift from the visual to the auditory representational activities, we can say that, in parallel with the RH specialization for face recognition, a greater role of the RH could also be observed in voice recognition. In particular, Van Lanker et al. [190] have shown that selective defects of voice recognition (phonoagnosia) are usually due to RH damage. In keeping with these data, Gainotti [191,192] showed, in two reviews of the perceptual and representational components of a familiar face, voice, and name recognition, that recognition of familiar faces and voices show a prevalent right lateralization and that the RH prevalence is greater in tasks involving familiar than unfamiliar faces and voices. These data suggest that the RH prominence in the recognition of faces and voices is not limited to their perceptual processing, but also extends to the domain of their cortical representations.

## 4. Concluding Remarks

Data discussed in previous sections of this review seem to show that different kinds of hemispheric asymmetries observed in patients with unilateral brain lesions could be subsumed by common mechanisms, more or less directly linked to the left lateralization of language. Among the processes more directly linked to the structuring influence of language on activities mediated by the LH, we can list: (1) the capacity to abstract away from perceptual similarities to generalize on the basis of categorical/conceptual criteria; (2) the capacity to solve, through covert language activities, complex non-verbal tasks that cannot be settled with a simple perceptual activity; and (3) the development of a conscious, propositional and controlled way of working, contrasting with the more concrete, automatic and context-dependent way of working of the RH. Among the hemispheric asymmetries linked to neuroplasticity processes resulting from the competition for cortical space within each hemisphere, we may include, on the contrary, the greater development of perceptual functions and of non-verbal representational activities in the RH. This mechanism assumes: (a) that the cortical areas involved in language processing in the left hemisphere had remained (at least in part) dedicated to the perceptual processing in the homologous areas of the RH and (b) that the left lateralization of language and the greater RH involvement in perceptual functions led to the development of two different ‘higher-order convergence zones’ [170,171], supporting respectively verbal and non-verbal forms of conceptual knowledge within the left and right hemispheres. Linked in part to the competition for cortical space and in part to the prevention of possible conflicts between the more primitive emotional and the more advanced cognitive systems could be the distinction between the left lateralization of language and the greater involvement of the RH in emotional functions. Even in this last domain, however, the contrast between the conscious vs. unconscious processing of emotional stimuli in the left vs. right hemisphere and the lack of disease awareness in patients with RH damage could be due to the automatic and poor conscious functioning of an RH neural system that cannot benefit from the structuring role of language.

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
