# Peer review of "Is There a Causal Link between the Left Lateralization of Language and Other Brain Asymmetries? A Review of Data Gathered in Patients with Focal Brain Lesions"

_brainsci, 2021, doi:10.3390/brainsci11121644_

Round 1
Reviewer 1 Report
This review paper discusses the relationship between left lateralization of language and hemispheric asymmetries in other brain functions, particularly based on data from patients with unilateral brain lesions. It reviewed a wide range of functions including some important findings, but there are several concerns as listed below.
・In the introduction, the author contrasts the causal hypothesis with the statistical hypothesis as a model for explaining left lateralization of language, but I think the data presented here from patients with unilateral brain lesion does not negate the possibility of the statistical hypothesis. A little more explanation for this point should be needed.
・A figure or table that summarizes the entire contents would be helpful for the readers to understand.
・The authors described “All these data confirm the difference between the automatic and unconscious processing of emotional information made by the non-verbal right hemisphere and the controlled and conscious processing of the same information made by the verbal left hemisphere. (lines 601-604)”. I think this statement is misleading and should be corrected, especially the contrast between unconscious processing by the right hemisphere and conscious processing by the left hemisphere. As for the response of the amygdala, this may be true, but it should not be interpreted as a difference between the left and right hemispheres of the brain. Indeed, as the author mentioned, the right insula contributes to conscious experience of emotion.
・Maybe it's because it relates to the previous point, but I do not follow the logic in section 3.3.2. I think that the evidence that patients with RH stroke sometimes showed anosognosia for hemiplegia suggests that RH contributes to conscious processing of our body or body schema. The author’s claim seems to be different from this, and I do not understand it.
・In section 3.1.6., please refer relevant papers which showed color perception can be affected by language (e.g. Davidoff, Davies, and Roberson, 1999).
・In some parts, the author described “greater involvement of the right hemisphere in more basic sensory-motor functions”. Evidence for greater involvement of the right hemisphere in sensory perception is described, but there is no evidence for motor functions. Rather, the author mentioned that the left hemisphere is important for voluntary execution of movement (section 3.2.2). Why did the author include the word "motor"?
Minor point
・The same sentence was written twice (lines 257 and 259).
・There are several typos in the text.
line 51: onducted should be conduncted
line 161: subststem -> subsystem
line 164:dealig -> dealing
line 711: revew ->review
line 784: predominatly -> predominantly 
Author Response
Reviewer 1
This review paper discusses the relationship between left lateralization of language and hemispheric asymmetries in other brain functions, particularly based on data from patients with unilateral brain lesions. It reviewed a wide range of functions including some important findings, but there are several concerns as listed below.
・In the introduction, the author contrasts the causal hypothesis with the statistical hypothesis as a model for explaining left lateralization of language, but I think the data presented here from patients with unilateral brain lesion does not negate the possibility of the statistical hypothesis. A little more explanation for this point should be needed.
Reply. A more detailed explanation of the predictions based on the causal and on the statistical hypothesis has been included on lines 44-56 of this revised version
・A figure or table that summarizes the entire contents would be helpful for the readers to understand.
・The authors described “All these data confirm the difference between the automatic and unconscious processing of emotional information made by the non-verbal right hemisphere and the controlled and conscious processing of the same information made by the verbal left hemisphere. (lines 601-604)”. I think this statement is misleading and should be corrected, especially the contrast between unconscious processing by the right hemisphere and conscious processing by the left hemisphere. As for the response of the amygdala, this may be true, but it should not be interpreted as a difference between the left and right hemispheres of the brain. Indeed, as the author mentioned, the right insula contributes to conscious experience of emotion.
Reply: In this revised version I have tried to develop with further arguments and references the hypothesis assuming that a contrast may exist between the unconscious processing by the right hemisphere and the conscious processing by the left hemisphere.
・Maybe it's because it relates to the previous point, but I do not follow the logic in section 3.3.2. I think that the evidence that patients with RH stroke sometimes showed anosognosia for hemiplegia suggests that RH contributes to conscious processing of our body or body schema. The author’s claim seems to be different from this, and I do not understand it.
Reply: In this revised version I have included on lines 655-689 some new references (and in particular the Daini’s [117] reworking of the Gazzaniga’s idea of the “Interpreter”, located in the left hemisphere). These new referencest clarify why I think that anosognosia of left-sided hemiplegia may not be due to damage of areas strictly involved in self-consciousness to a “disconnection” between what is processed in the right hemisphere and the self-consciousness-related system in the left hemisphere
・In section 3.1.6., please refer relevant papers which showed color perception can be affected by language (e.g. Davidoff, Davies, and Roberson, 1999).
Reply: I did not included papers such as that suggested by the reviewer because section 3.1.6. is part of a more general section dealing with Data obtained studying the relations between verbal and non-verbal disorders of aphasic patients, whereas the Davidoff et al. (1999) paper deals with cross-cultural perception and cognition studies. I have, however, cited on lines 381-389 of this revised version a further study investigating color perception and categorization in aphasic patients
・In some parts, the author described “greater involvement of the right hemisphere in more basic sensory-motor functions”. Evidence for greater involvement of the right hemisphere in sensory perception is described, but there is no evidence for motor functions. Rather, the author mentioned that the left hemisphere is important for voluntary execution of movement (section 3.2.2). Why did the author include the word "motor"?
Reply: In this revised version have changed everywhere in the text sensory-motor into sensory-perceptual
Minor point
・The same sentence was written twice (lines 257 and 259).
・There are several typos in the text.
line 51: onducted should be conduncted
line 161: subststem -> subsystem
line 164:dealig -> dealing
line 711: revew ->review
line 784: predominatly -> predominantly
Reply: I thank the reviewer for this precious editing
Reviewer 2 Report
In general, I found some of the concepts from the linguistic approach well posed, and the perspective itself quite interesting (and/or complementary to the neuroscience perspective).
I have noticed that: In a short section "3.2.2.. The selective disruption of the voluntary execution of movements in left brain-damaged patients with apraxia" the Author talks about praxis-language links. This is done mainly in the context rather old literature (with the newest reference from 2015). Perhaps the contributing Author could check if any useful additional information for this topic -i.e., the laterality of mechanisms for language and praxis production - could be found in the following reference: Kroliczak, G., Buchwald, M., Kleka, P., Klichowski, M., Potok, W., Nowik, A. M., Randerath, J., & Piper, B. J. (2021). Manual praxis and language-production networks, and their links to handedness. Cortex, 140, 110-127. https://doi.org/10.1016/j.cortex.2021.03.022
In this work (open access), the latest developments in the field of praxis-language links are discussed and tested, and at least citing it could be very beneficial for readers particularly interested in this area.
Minor comment related to the text itself: I noticed that two sentences or so were (copied and) pasted twice (next to each other).
Author Response
In general, I found some of the concepts from the linguistic approach well posed, and the perspective itself quite interesting (and/or complementary to the neuroscience perspective).
Reply: I thank the reviewer for the positive evaluation of my work
I have noticed that: In a short section "3.2.2.. The selective disruption of the voluntary execution of movements in left brain-damaged patients with apraxia" the Author talks about praxis-language links. This is done mainly in the context rather old literature (with the newest reference from 2015). Perhaps the contributing Author could check if any useful additional information for this topic -i.e., the laterality of mechanisms for language and praxis production - could be found in the following reference: Kroliczak, G., Buchwald, M., Kleka, P., Klichowski, M., Potok, W., Nowik, A. M., Randerath, J., & Piper, B. J. (2021). Manual praxis and language-production networks, and their links to handedness. Cortex, 140, 110-127. https://doi.org/10.1016/j.cortex.2021.03.022
In this work (open access), the latest developments in the field of praxis-language links are discussed and tested, and at least citing it could be very beneficial for readers particularly interested in this area.
Reply: I have included on lines 505-518 of this revised version a paragraph and some references dealing with these latest developments in the field of praxis-language links.
Minor comment related to the text itself: I noticed that two sentences or so were (copied and) pasted twice (next to each other).
Reply: I thank the reviewer for this precious editing work 
Reviewer 3 Report
The article “Is there a causal link between the left lateralization of language 2 and other brain asymmetries? A review of data gathered in pa-3 tients with focal brain lesions.” by Gainotti provides a very comprehensive and interesting overview of the literature on the relation between unilateral brain lesions and how they affect lateralized brain functions with a specific focus on language lateralization. The review is very well written and I have only very few (minor) comments.
- My only content-related question relates to the first part of the review. While reading, I wondered whether it would be worthwhile to add working memory as a topic to this first section. According to Baddeley’s model, there should be differences with regard to working memory performance in patient’s with unilateral brain lesions and these deficits should likely be more in the phonological loop domain if left-sided and in the visuospatial sketchpad domain if right-sided. I wonder if the author knows of studies that support these assumptions. If so, I believe that they could add another layer to the manuscript given that working memory is such a critical cognitive domain that is essential for many executive functions.
- There seem to be some inconsistencies on the section headings and how they are presented. Sometimes they are underlined, sometimes not (e.g. 3.3.1 and 3.3.2) and they could also sometimes be more on point (e.g. 3.1.6).
- They are a number of typos in the manuscript. L30, that instead of than. L51: onducted instead of conducted. L121: dot missing. L183: dot missing. Line 185: two a’s. L186: this section instead of sections. L214: this reason instead of reasons. L216: dot missing. L234: pioneers instead of pionners. L257: the sentence comes twice. L345,403 and 416: the “the” before the author names feels unusual to me. L492: explicitly instead of explicitely. L625: In the title, USN is capitalized, but not in the text. This seems a bit inconsistent. L711: review instead of revew. L719 and 746: I would speak of healthy instead of normal since normal is rather ill-defined. L749: possible instead of posible.
Author Response
The article “Is there a causal link between the left lateralization of language 2 and other brain asymmetries? A review of data gathered in patients with focal brain lesions.” by Gainotti provides a very comprehensive and interesting overview of the literature on the relation between unilateral brain lesions and how they affect lateralized brain functions with a specific focus on language lateralization. The review is very well written and I have only very few (minor) comments.
Reply: I am very grateful to the reviewer for the very positive evaluation of my work.
1.My only content-related question relates to the first part of the review. While reading, I wondered whether it would be worthwhile to add working memory as a topic to this first section. According to Baddeley’s model, there should be differences with regard to working memory performance in patient’s with unilateral brain lesions and these deficits should likely be more in the phonological loop domain if left-sided and in the visuospatial sketchpad domain if right-sided. I wonder if the author knows of studies that support these assumptions. If so, I believe that they could add another layer to the manuscript given that working memory is such a critical cognitive domain that is essential for many executive functions.
Reply: I have included on lines 862-875 of this revised version a short paragraph and a couple of references which support Baddley’s model (and the similar Kosslyn’s model)
1.There seem to be some inconsistencies on the section headings and how they are presented. Sometimes they are underlined, sometimes not (e.g. 3.3.1 and 3.3.2) and they could also sometimes be more on point (e.g. 3.1.6).
Reply: In this revised version I have tried to be more consistent on the section headings.
1.They are a number of typos in the manuscript. L30, that instead of than. L51: onducted instead of conducted. L121: dot missing. L183: dot missing. Line 185: two a’s. L186: this section instead of sections. L214: this reason instead of reasons. L216: dot missing. L234: pioneers instead of pionners. L257: the sentence comes twice. L345,403 and 416: the “the” before the author names feels unusual to me. L492: explicitly instead of explicitely. L625: In the title, USN is capitalized, but not in the text. This seems a bit inconsistent. L711: review instead of revew. L719 and 746: I would speak of healthy instead of normal since normal is rather ill-defined. L749: possible instead of posible.
Reply: I thank the reviewer for the precious editing work.
Round 2
Reviewer 1 Report
The author has adequately addressed all the points I raised and the revision has been greatly improved.